# Impacts of type 2 diabetes mellitus and hypertension on the incidence of cardiovascular diseases and stroke in China real-world setting: a retrospective cohort study

Yan Liu,[1] Jie Li,[1] Ying Dou,[2] Hongshan Ma 🔟 [3]

¹Department of Endocrinology, The Third People's Hospital of Datong, Datong, China
²Department of Medicine, Ashermed Pharmaceutical Technology Co Ltd, Shanghai, China
³Department of Cardiology, The Third People's Hospital of Datong, Datong, China

**Correspondence to**
Hongshan Ma;
mahongshannew@163.com

## ABSTRACT

**Objective** The prevalence of type 2 diabetes mellitus (T2DM) and hypertension (HTN) has notably increased in recent years. However, there is little evidence from large-scale studies assessing the joint effect of T2DM and HTN on the risk of cardiovascular events in China. This study was performed to investigate the association of T2DM and HTN with the incidence of combined vascular events (VEs) and stroke in China.

**Design** A retrospective cohort study.

**Setting** Data were collected from the SuValue database which includes the electronic medical records of >90 million patients from 161 hospitals across 18 provinces in China.

**Participants** Patients aged ≥18 with a diagnosis of T2DM and/or HTN were included. Non-T2DM and non-HTN patients were included in this study as controls.

**Outcomes** Incidence of combined VEs and stroke during the study.

**Results** In the current study, 8012 patients with T2DM, 9653 patients with HTN, 3592 patients with both T2DM and HTN and 10 561 patients without T2DM or HTN were included. T2DM was significantly associated with combined VE and stroke risk (HR 1.332, 95% CI 1.134 to 1.565 and HR 1.584, 95% CI 1.246 to 2.014, respectively). HTN was significantly associated with combined VE and stroke risk (HR 3.244, 95% CI 2.946 to 3.572 and HR 4.543, 95% CI 3.918 to 5.268, respectively). T2DM combined with HTN was significantly associated with combined VE and stroke risk (HR 3.002, 95% CI 2.577 to 3.497 and HR 4.151, 95% CI 3.346 to 5.149, respectively). HTN was associated with a higher combined VE and stroke risk than T2DM (HR 2.435, 95% CI 2.113 to 2.805 and HR 2.868, 95% CI 2.341 to 3.513, respectively).

**Conclusion** T2DM and HTN were strongly associated with combined VE and stroke risk; however, the HTN-only group had a higher combined VE and stroke risk than the T2DM-only group.

## Strengths and limitations of this study

► This study was designed to analyse the electronic medical records of patients in a real-world setting.
► Cardiovascular risk factors were collected in this study.
► Body Mass Index and lifestyle factors, such as smoking and alcohol consumption, were not recorded in the electronic medical records of patients.
► Mortality data were not accessible through the electronic medical records of patients.

over two-thirds of patients with diabetes.[3 4] The development of HTN coincides with the development of hyperglycaemia.[3] Insulin resistance and hyperinsulinaemia might promote atherogenesis, thereby affecting blood pressure homeostasis.[5 6]

Diabetes has been reported to be a strong risk factor for cardiovascular disease, all-cause mortality, coronary heart disease, ischaemic heart disease and stroke in many studies.[7–11] According to the Framingham Heart Study, adults with diabetes had an absolute twofold increased risk of cardiovascular disease compared with subjects without diabetes.[12] The total cardiovascular disease burden in diabetes patients has increased throughout the past four decades.[3] A cross-sectional study showed that HTN was common in patients with newly diagnosed diabetes, and HTN patients had a higher prevalence of cardiovascular events than normotensive subjects before the diagnosis of diabetes.[13] HTN has also been reported to be one of the strongest risk factors for cardiovascular disease, including coronary disease, vascular heart disease and cerebral stroke.[14–20] Adults with coexistent diabetes and HTN dramatically increased the risk of cardiovascular disease by two to four times compared with adults

## INTRODUCTION

The prevalence of type 2 diabetes mellitus (T2DM) increased to 10.4% in 2013 and 11.2% in 2015 from 0.67% in 1980 in China.[1 2] Hypertension (HTN) is found in

**Table 1** Baseline characteristics of the included patients

| | HTN-only | T2DM-only | T2DM and HTN | Non-T2DM and non-HTN | P value |
|---|---|---|---|---|---|
| **Total** | 9653 | 8012 | 3592 | 10 561 | |
| Sex | | | | | |
| Male | 4419 (45.8%) | 4006 (50.0%) | 1634 (45.5%) | 4889 (46.3%) | – |
| Female | 5234 (54.2%) | 4006 (50.0%) | 1958 (54.5%) | 5672 (53.7%) | – |
| Age | | | | | |
| 18–29 (n, %) | 79 (0.8%) | 204 (2.5%) | 31 (0.9%) | 1372 (13.0%) | – |
| 30–39 (n, %) | 602 (6.2%) | 831 (10.4%) | 139 (3.9%) | 2439 (23.1%) | – |
| 40–49 (n, %) | 1856 (19.2%) | 2323 (29.0%) | 553 (15.4%) | 3227 (30.6%) | – |
| 50–59 (n, %) | 1856 (20.7%) | 2297 (28.7%) | 927 (25.8%) | 1954 (18.5%) | – |
| 60–69 (n, %) | 2552 (26.4%) | 1704 (21.3%) | 1194 (33.2%) | 1179 (11.2%) | – |
| 70–79 (n, %) | 1715 (17.8%) | 556 (6.9%) | 573 (16.0%) | 336 (3.2%) | – |
| ≥80 (n, %) | 852 (8.8%) | 97 (1.2%) | 175 (4.9%) | 54 (0.5%) | – |
| Triglyceride (mmol/L) | 1.86 (1.54) | 2.35 (2.55) | 2.29 (1.85) | 1.71 (1.65) | <0.0001 |
| Total cholesterol (mmol/L) | 5.31 (1.19) | 5.32 (1.41) | 5.29 (1.28) | 5.28 (1.15) | 0.3506 |
| HDL-C (mmol/L) | 1.41 (0.41) | 1.25 (0.45) | 1.26 (0.39) | 1.40 (0.39) | <0.0001 |
| LDL-C (mmol/L) | 3.17 (1.00) | 3.07 (1.03) | 3.02 (0.98) | 3.03 (0.90) | <0.0001 |
| Serum creatinine (µmol/L) | 81.11 (39.35) | 70.57 (23.94) | 84.26 (59.38) | 76.78 (20.42) | <0.0001 |
| HbA1c (%) | 5.58 (1.38) | 9.41 (3.11) | 8.41 (2.63) | 5.73 (1.31) | <0.0001 |
| Fasting blood-glucose (mmol/L) | 5.76 (1.28) | 11.15 (5.28) | 9.88 (5.14) | 5.58 (1.13) | <0.0001 |
| Serum insulin (pmol/litre) | 95.63 (60.05) | 117.64 (288.36) | 78.43 (57.89) | 9.98 (3.21) | 0.6502 |
| Proteinuria (positive) | 310 (3.2%) | 145 (1.8%) | 126 (3.5%) | 143 (1.4%) | <0.0001 |
| Diabetes complications | | | | | |
| Diabetic nephropathy | – | 450 (5.6%) | 446 (12.4%) | – | – |
| Diabetic retinopathy | – | 313 (3.9%) | 159 (4.4%) | – | – |
| Diabetic neuropathy | – | 462 (5.8%) | 225 (6.3%) | – | – |
| Diabetic lower limb vascular disease | – | 7 (0.09%) | 15 (0.42%) | – | – |
| Diabetic foot | – | 80 (1.0%) | 61 (1.7%) | – | – |

For continuous variables, data were presented as mean (SD) and p values for four groups were calculated using analysis of variance test. For categorical variables, data were presented as number (frequency) and p values for four groups were calculated using $\chi^2$ test.
HDL-C, high-density lipoprotein cholesterol; HTN, hypertension; LDL-C, low-density lipoprotein cholesterol; T2DM, type 2 diabetes mellitus.

without HTN or diabetes.[3] Thus, diabetes and HTN are thought to be poor companions, and blood pressure control is critical in diabetes patients with HTN. In addition, a study demonstrated that HTN had a stronger association with atherosclerotic cardiovascular disease than diabetes.[21]

The effects of HTN and diabetes on the risk of cardiovascular disease have been investigated in American, Finnish, Japanese and Iranian populations.[21–24] However, there is little evidence from large-scale studies assessing the joint effect of HTN and diabetes on the risk of cardiovascular events in China. The purpose of the study was to evaluate the impact of HTN and T2DM on the risk of cardiovascular disease and stroke in Chinese adults using the SuValue database.

## METHODS
### Study design
The SuValue database is a big-data hospital information system (HIS) database that includes data on >90 million patients from 161 hospitals across 18 provinces in China.[25] This was a retrospective cohort study designed to evaluate the risk of cardiovascular diseases and stroke in patients with T2DM and/or HTN from 2004 to 2015 in a real-world Chinese setting. Patients were included if they met the following criteria: (1) aged ≥18; (2) newly diagnosed T2DM and/or HTN; (3) had baseline examination records before or within 3 months at the first diagnosis (details of baseline examination were described in the Baseline Parameters section) and (4) electronic medical records (EMRs) could be found 1 year later after

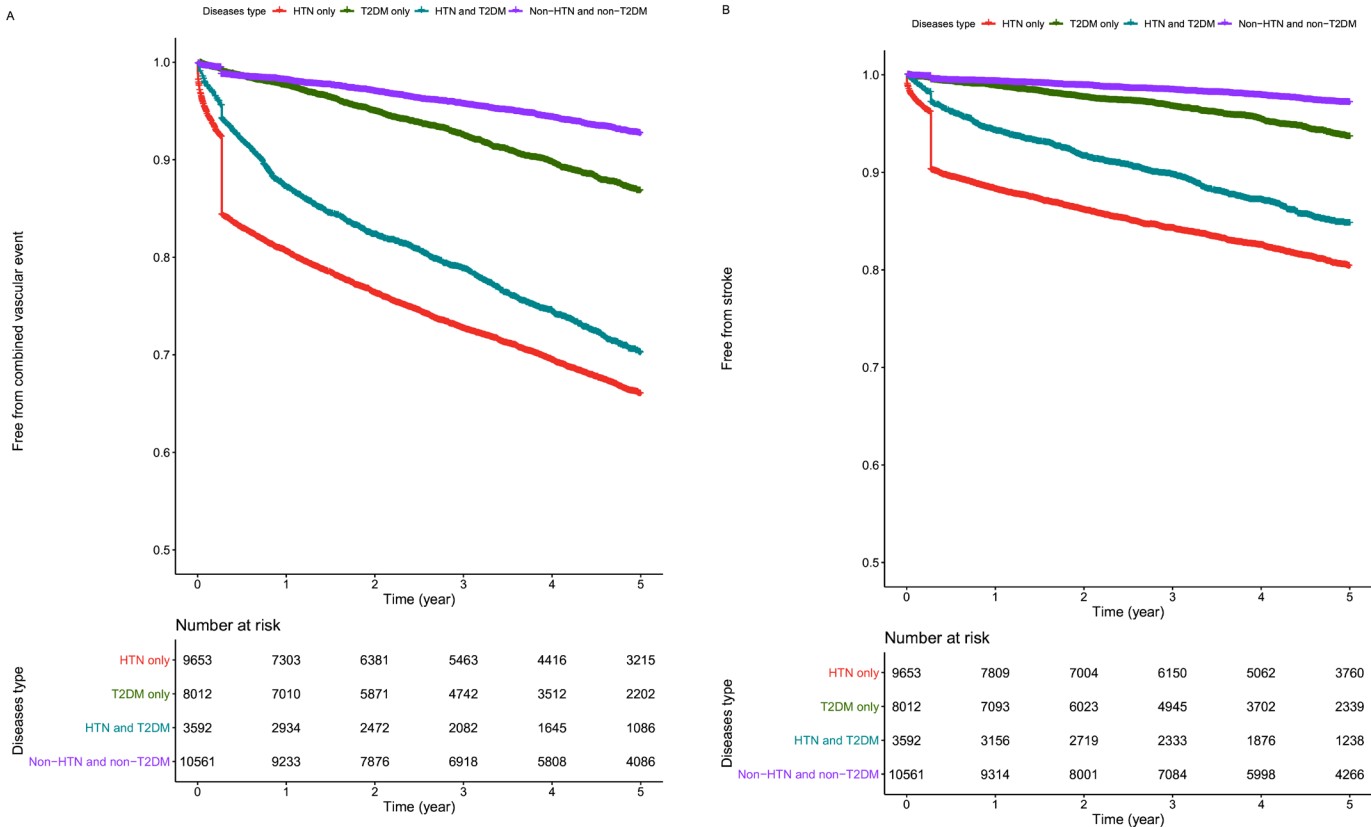

**Figure 1** Kaplan–Meier survival curve of combined vascular events (A) and stroke (B) among different groups. HTN, hypertension; T2DM, type 2 diabetes mellitus.

the first diagnosis of T2DM and/or HTN. Patients were excluded as follows: missing sex information; had been diagnosed with stroke, myocardial infarction, coronary heart disease, heart failure; had received coronary artery bypass grafting or percutaneous coronary intervention before the first diagnosis of T2DM and/or HTN and had abnormal kidney function (normal range for serum creatinine: 54–106 µmol/L for men; 44–97 µmol/L for women). We also included non-T2DM and non-HTN patients who underwent the baseline examination but were not from the obstetrics and gynaecology department, cancer department, neurology department or cardiology department. Authorisation for the SuValue database was obtained when the database was set up, so neither ethics review nor written informed patient consent were needed for this analysis. In addition, all patients' EMRs were deidentified and anonymised when the SuValue database was constructed.

### Baseline parameters

Sex, age, triglyceride, total cholesterol, high-density lipoprotein cholesterol (HDL-C), low-density lipoprotein cholesterol (LDL-C), serum creatinine, fasting blood glucose, serum insulin, proteinuria and medication (antidiabetics and antihypertensive medications) at baseline were captured from EMRs before or within 3 months of the first diagnosis of T2DM and/or HTN.

### OUTCOMES

We examined two outcomes of interest: combined vascular events (VEs) and stroke. Combined VEs include stroke, myocardial infarction, coronary heart disease, heart failure and coronary bypass and percutaneous coronary intervention. The outcomes were defined as the first event or last record before 31 December 2019 among the event-free cases according to the diagnosis in the patients' EMRs.

### Statistical analyses

Categorical variables were described using frequencies and percentages. Continuous variables were described using the mean (SD) if normally distributed or as the median IQR if not. Baseline characteristics were compared using analysis of variance or $\chi^2$ tests. Kaplan–Meier survival analysis of the incidence of combined VEs and stroke according to the presence of T2DM and/or HTN was performed. A Cox proportional hazards model was used to assess the association between diseases and each outcome, which was adjusted for cardiovascular risk factors, including sex, age, triglycerides, total cholesterol, HDL-C and LDL-C. Unadjusted, sex-adjusted and age-adjusted and cardiovascular risk factor-adjusted HRs were calculated. A p<0.05 (two-sided) was considered statistically significant. All statistical analyses were performed using SAS V.9.4 (SAS Institute, Cary, NC, USA).

**Table 2** Association of type 2 diabetes mellitus (T2DM) and/or hypertension (HTN) with combined vascular events and stroke compared with the non-T2DM and non-HTN

| | Combined vascular event (VE) | | Stroke | |
|---|---|---|---|---|
| | HR (95% CI) | P value | HR (95% CI) | P value |
| Unadjusted | | | | |
| Non-T2DM and non-HTN | 1 (reference) | | 1 (reference) | |
| T2DM-only | 1.747 (1.566 to 1.949) | <0.0001 | 2.077 (1.755 to 2.459) | <0.0001 |
| HTN-only | 6.246 (5.712 to 6.830) | <0.0001 | 8.642 (7.517 to 9.934) | <0.0001 |
| T2DM and HTN | 4.930 (4.93 to 5.474) | <0.0001 | 5.990 (5.102 to 7.032) | <0.0001 |
| Age-adjusted and sex-adjusted | | | | |
| Non-T2DM and non-HTN | 1 (reference) | | 1 (reference) | |
| T2DM-only | 1.258 (1.127 to 1.405) | <0.0001 | 1.464 (1.235 to 1.735) | <0.0001 |
| HTN-only | 3.344 (3.046 to 3.670) | <0.0001 | 4.409 (3.817 to 5.093) | <0.0001 |
| T2DM and HTN | 2.595 (2.329 to 2.892) | <0.0001 | 3.021 (2.561 to 3.563) | <0.0001 |
| Risk factors-adjusted | | | | |
| Non-T2DM and non-HTN | 1 (reference) | | 1 (reference) | |
| T2DM-only | 1.332 (1.134 to 1.565) | 0.0005 | 1.584 (1.246 to 2.014) | <0.0001 |
| HTN-only | 3.244 (2.946 to 3.572) | <0.0001 | 4.543 (3.918 to 5.268) | <0.0001 |
| T2DM and HTN | 3.002 (2.577 to 3.497) | <0.0001 | 4.151 (3.346 to 5.149) | <0.0001 |

Risk factors include age, sex, triglycerides, total cholesterol, high-density lipoprotein cholesterol, low-density lipoprotein cholesterol.
HTN, hypertension; T2DM, type 2 diabetes mellitus; VE, combined vascular event.

## Public and patient involvement

Neither patients nor the public were involved in the design, conduct, reporting or dissemination plans of this research.

## RESULTS

In the current study, 8012 patients with T2DM-only, 9653 patients with HTN-only and 3592 patients with both T2DM and HTN and 10 561 patients without T2DM or HTN were selected from the SuValue database. The median follow-up duration was 4.1 (2.4, 4.9) years. The general characteristics of the study population at baseline are presented in table 1. Comparing baseline characteristics between the four groups revealed significant differences, except in total cholesterol (p=0.3506) and serum insulin (p=0.6502).

Kaplan–Meier analyses demonstrated increased combined VE and stroke risk in the HTN group (figure 1). In the unadjusted models, compared with non-HTN and non-T2DM, T2DM had an HR of 1.747 for combined VEs (95% CI 1.566 to 1.949, p<0.0001) and 2.077 for stroke (95% CI 1.755 to 2.459, p<0.0001). Further adjustment with age and sex and then with cardiovascular risk factors attenuated the association of T2DM with combined VE and stroke risk (HR 1.332, 95% CI 1.134 to 1.565 and HR 1.584, 95% CI 1.246 to 2.014, respectively; p<0.001, table 2).

Kaplan–Meier analyses demonstrated an increased combined VE and stroke risk in the HTN group (figure 1). The HTN-only group had a more than 6-fold increased risk

of combined VE (95% CI 5.712 to 6.830) and a more than eightfold increased risk of stroke (95% CI 7.517 to 9.934) in unadjusted models (all Ps <0.0001). These associations were attenuated but still significant after adjustment for age and sex and then for other major cardiovascular risk factors for both combined VE and stroke risk (HR 3.244, 95% CI 2.946 to 3.572 and HR 4.543, 95% CI 3.918 to 5.268, respectively; all Ps <0.0001, table 2).

Kaplan–Meier analyses demonstrated increased combined VE and stroke risk in the HTN and T2DM groups (figure 1). For combined VEs, the HR of the HTN and T2DM group was 4.93 (95% CI 4.93 to 5.474). The HTN and T2DM groups had an increased risk of stroke to 5.990 (95% CI 5.102 to 7.032) in the unadjusted models (all Ps <0.0001). These associations were attenuated but still significant after adjustment for age and sex and then other major cardiovascular risk factors for both combined VE risk and stroke risk (HR 3.002, 95% CI 2.577 to 3.497 and HR 4.151, 95% CI 3.346 to 5.149, respectively; all Ps <0.0001, table 2).

Compared with the HTN-only group, the unadjusted HRs for combined VE risk were 0.789 and 0.693 for stroke risk in both the T2DM and HTN groups (all Ps <0.0001). However, after adjustment for major cardiovascular risk factors, there was no significantly reduced risk for either the T2DM or HTN group compared with the HTN-only group (table 3).

In the unadjusted model, the HTN and T2DM group and the HTN-only group had a higher combined VE and stroke risk than the T2DM-only group. After adjustment

**Table 3** Association of type 2 diabetes mellitus (T2DM) and hypertension (HTN) with combined vascular events and stroke compared with HTN-only

| | Combined vascular event (VE) | | Stroke | |
|---|---|---|---|---|
| | HR (95% CI) | P value | HR (95% CI) | P value |
| Unadjusted | | | | |
| HTN-only | 1 (reference) | | 1 (reference) | |
| T2DM and HTN | 0.789 (0.732 to 0.851) | <0.0001 | 0.693 (0.624 to 0.769) | <0.0001 |
| Age-adjusted and sex-adjusted | | | | |
| HTN-only | 1 (reference) | | 1 (reference) | |
| T2DM and HTN | 0.776 (0.720 to 0.836) | <0.0001 | 0.685 (0.617 to 0.761) | <0.0001 |
| Risk factors-adjusted | | | | |
| HTN-only | 1 (reference) | | 1 (reference) | |
| T2DM and HTN | 0.925 (0.814 to 1.052) | 0.2352 | 0.914 (0.771 to 1.082) | 0.2959 |

Risk factors include age, sex, triglycerides, total cholesterol, high-density lipoprotein cholesterol, low-density lipoprotein cholesterol.
HTN, hypertension; T2DM, type 2 diabetes mellitus; VE, combined vascular event.

for age and sex and major cardiovascular risk factors, both the HTN and T2DM groups and the HTN-only group were still significantly associated with combined VE and stroke risk (table 4).

## DISCUSSION

In the present study, having HTN and/or T2DM was significantly associated with combined VE and stroke before and after adjustment for major cardiovascular risk factors compared with the non-T2DM and non-HTN. The association of T2DM with the risk of both combined VE and stroke attenuated after adjusting for major

cardiovascular risk factors. In addition, the HTN-only group had a higher combined VE and stroke risk than the T2DM-only group.

To our knowledge, our study is the first to investigate the combined effect of HTN and T2DM in a large cohort in a real-world setting in Chinese patients. Several studies investigating the different impacts of HTN and diabetes on cardiovascular disease incidence and mortality have been conducted in Finland, America and Japan.[21–23 26] HTN and/or T2DM were associated with an increased risk of combined VE and stroke in this study. The HR for cardiovascular disease was approximately two in

**Table 4** Association of type 2 diabetes mellitus (T2DM) and/or hypertension (HTN) with combined vascular events and stroke compared with T2DM-only

| | Combined vascular event (VE) | | Stroke | |
|---|---|---|---|---|
| | HR (95% CI) | P value | HR (95% CI) | P value |
| Unadjusted | | | | |
| T2DM-only | 1 (reference) | | 1 (reference) | |
| HTN-only | 3.575 (3.296 to 3.877) | <0.0001 | 4.160 (3.701 to 4.675) | <0.0001 |
| T2DM and HTN | 2.821 (2.558 to 3.112) | <0.0001 | 2.883 (2.504 to 3.321) | <0.0001 |
| Age-adjusted and sex-adjusted | | | | |
| T2DM-only | 1 (reference) | | 1 (reference) | |
| HTN-only | 2.657 (2.447 to 2.885) | <0.0001 | 3.011 (2.676 to 3.389) | <0.0001 |
| T2DM and HTN | 2.062 (1.868 to 2.276) | <0.0001 | 2.063 (1.789 to 2.379) | <0.0001 |
| Risk factors-adjusted | | | | |
| T2DM-only | 1 (reference) | | 1 (reference) | |
| HTN-only | 2.435 (2.113 to 2.805) | <0.0001 | 2.868 (2.341 to 3.513) | <0.0001 |
| T2DM and HTN | 2.253 (1.876 to 2.706) | <0.0001 | 2.620 (2.031 to 3.380) | <0.0001 |

Risk factors include age, sex, triglycerides, total cholesterol, high-density lipoprotein cholesterol and low-density lipoprotein cholesterol.
.HTN, hypertension; T2DM, type 2 diabetes mellitus; VE, combined vascular event.

the T2DM-only group, which was lower than the three reported in the Framingham cohort of 1952–1974.[11] Both HTN and T2DM have been shown to be strong risk factors for cardiovascular disease mortality.[3]

In unadjusted analyses and adjustment models for major cardiovascular risk factors, we observed that patients with both T2DM and HTN showed an increased risk of combined VE and stroke compared with those with T2DM alone. These results suggest that HTN confers an enhanced risk of cardiovascular disease. Similarly, the coexistence of HTN and T2DM conferred an increased risk of cardiovascular disease incidence compared with T2DM-only.[3 27]

After adjustment for major cardiovascular risk factors, patients with T2DM and HTN did not have a significantly increased combined VE and stroke risk compared with patients with HTN-only. In this study, we included patients with HTN and/or T2DM who had medication records. Antihypertension treatment, such as ACE inhibitors, calcium-channel blockers, or β blockers, has been shown to reduce the risk of cardiovascular diseases according to several randomised trials and meta-analysis studies.[28 29] A meta-analysis demonstrated that antihypertensive treatment decreased the risk of cardiovascular disease and stroke among patients with a history of cardiovascular disease without HTN.[30] Thus, combined T2DM and HTN did not increase the risk of combined VE and stroke compared with HTN, which may be due to the antihypertensive treatment in this population.

In addition, we also observed that the HTN-only group was more associated with combined VE and stroke risk than the T2DM group. The results were similar to a previous report that HTN alone was more related to all-cause and atherosclerotic cardiovascular disease than T2DM alone in community-dwelling older adults.[21] A study performed in older Iranian adults showed that T2DM alone increased the coronary heart disease incidence, stroke incidence and all-cause mortality compared with HTN alone.[24] Thus, a prospective study is necessary for further analysis.

The current study has several significant strengths. First, this study included a large number of patients. Second, data about cardiovascular risk factors were collected in this study. However, there were several limitations in this study. First, Body Mass Index and lifestyle factors, such as smoking and alcohol consumption, were not recorded in the EMRs. Second, mortality data could not be accessed through the HIS system of hospitals.

## CONCLUSIONS

In summary, HTN and/or T2DM was strongly associated with an increased risk of combined VE and stroke independent of conventional cardiovascular risk factors. T2DM does not seem to be a risk factor for combined VE and stroke in HTN patients after adjusting for major cardiovascular risk factors. HTN had a significantly higher combined VE and stroke risk than T2DM. However, a prospective study investigating the impact of HTN and/or T2DM on combined VE and stroke risk needs to be performed.

**Contributors** HM and YL designed the study, analysed the data and wrote the first draft of the manuscript. HM, YL, JL and YD revised it critically for important intellectual content and approved the final version. HM accepts full responsibility for the finished work and/or the conduct of the study, had access to the data, and controlled the decision to publish.

**Funding** The authors have not declared a specific grant for this research from any funding agency in the public, commercial or not-for-profit sectors.

**Competing interests** None declared.

**Patient and public involvement** Patients and/or the public were not involved in the design, or conduct, or reporting or dissemination plans of this research.

**Patient consent for publication** Not required.

**Ethics approval** All analyses were performed based on the SuValue database. Authorisation for the SuValue database was obtained when the database was set up, so neither ethics review or written informed patient consent were needed for this analysis. In addition, all patients' electronic medical records (EMRs) were deidentified and anonymised when the SuValue database was constructed.

**Provenance and peer review** Not commissioned; externally peer reviewed.

**Data availability statement** Data may be obtained from a third party and are not publicly available. All data relevant to the study are included in the article were obtained from SuValue database which belongs to a third party and is not publicly available.

**ORCID iD**
Hongshan Ma http://orcid.org/0000-0002-4029-4774

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
