## [Reviewer comments · BMJ Open]

ARTICLE DETAILS

TITLE (PROVISIONAL)	Impacts of type 2 diabetes mellitus and hypertension on the incidence of cardiovascular diseases and stroke in China real-world setting: A retrospective cohort study
AUTHORS	Liu, Yan; Li, Jie; Dou, Ying; Ma, Hongshan

VERSION 1 – REVIEW

REVIEWER	Debele, Gebiso Roba Mettu University, Public health
REVIEW RETURNED	16-Aug-2021

GENERAL COMMENTS	Yan Liu, Jie Li, Ying Dou and Hongshan Ma; performed a Cohort study, in order to assess the Impacts of type 2 diabetes mellitus and hypertension on cardiovascular disease and stroke in Chinese patients. The paper is quite original if related to the geographic area. The methodology is correct. The results support the conclusions. However, this reviewer raises some minor criticisms that have to be addressed by the authors and presented as follows: General Comments Abstract Introduction: Please add the gaps of your study before your expectation Methods: Better to add the study design and analysis method Finding: "Findings For the current study, 8,012 patients with T2DM, 9,653 patients with HTN and 3,592 patients with both T2DM" and what?? Better to replace the word "Interpretation" by Conclusion in the abstract. Methods "We examined 3 outcomes of interest" but what exists on your manuscript is only 2 outcomes which is VE and stroke. What is your third outcome? Where is the operational definition of your outcome? Results "Prevalent HTN was significantly associated with a 524.6% increased risk of VE (-95% CI 5.712-6.830) and 764.2% increased risk of stroke". What does mean by 524.6% and 764.2% percent? And you need to clarify all those percent's above 100%. Some of your references was too old and you need to update.
---

REVIEWER	Chiwanga, Faraja Muhimbili National Hospital, Endocrinology and Diabetes
REVIEW RETURNED	21-Aug-2021

GENERAL COMMENTS	General comments: 1) There are several typographical errors throughout the manuscript which makes it difficult to follow arguments the authors are making 2) This study has been described as cross-sectional however there are several facts that indicate it is not a cross-sectional type of study, like; i)excluding patients with outcome of interest and ii)prospectively following up patients to observe occurrence of interest. It is therefore difficult to focus on the results and analysis without first understanding the method the authors used.
--

REVIEWER	Iannaccone, Mario Ospedale San Giovanni Bosco, Cardioly
REVIEW RETURNED	24-Aug-2021

GENERAL COMMENTS	The authors should be congratulated for their efforts, in particular they collected a big amount of data. However the paper suffers of important limitation, such as the lack of data about BMI, further the endpoint definition is really unclear (in particular for the composite endpoint). Indeed the results/conclusion mainly confirm previous known experience such as the FRAMINGAM study cohort and similar epidemiological studies.
---

REVIEWER	Barrett, Sheila Northern Illinois University, Department of Nutrition, Dietetics and Hospitality Administration
REVIEW RETURNED	28-Aug-2021

GENERAL COMMENTS	General- some typos and grammatical mistakes noted. Need to left justify to correct the additional spaces in the document. Ned to use the past tense, the study has already been completed. Statistical analyses were rigorous but in the results 3 sentences started off with Kaplan-Meier Analyses, change these beginning sentences. Abstract Some typos noted; Line 9- change was largely increased to “has” largely increased. The study was performed “to” “investigate” Methods- identify the type of study in the abstract, seems like a retrospective cross-sectional study. Line 22- is confusing, if the study is retrospective, why mention follow -up for 5 years? That sounds like a prospective cohort study. Line 27- add HTN after T2DM Interpretation – this should be conclusion? Rewrite- “Subjects with T2DM and HTN were strongly associated with VE and stroke risk, however, the HTN only group was more strongly associated with VE and stroke risk compared with the T2DM only group. Page 4- strengths and Limitations- separate the strengths from the limitations. Line 17- reword “Data on cardiovascular risk factors... Lines 28- 34- need a rewrite- Suggestions “We included patients who were admitted to this hospital for the first time but it is unclear if the patient was first diagnosed as type 2 diabetes mellites or hypertension which increased the heterogeneity of the study population. “Not clear what is meant on the bullet point for
---

	mortality and BMI, are these strengths or limitations? Explain to make them clearer as well as separate strengths from limitations. Introduction Line 28- cardiovascular diseases “compared” with Line 35- study showed HTN is commonly “found” in ... Line 38- add the word “who” after HTN patients Line 54- edit comparing to “compared” (You are mixing up tenses, present continuous with past tense), the study is over, so use past tense. Lines 14-20- fix the purpose of the study. Suggestions “ The purpose of the study was to evaluate the impact of HTN and t2DM on cardiovascular diseases risk and stroke in Chinese adults using the SuValue database. Methods Page 6- Line 33- add “events/diseases after the risk of cardiovascular... Page 7- line 17- change was to were. Page 7- Line 46- remove the superscript “st’ after December 31, superscript is typically used before the month, should remain as December 31, 2019. Or 31st December 2019. Results Page 8- line 28- add “the” in front of SuValue Page 8- line 36- add the p values in parentheses - ... for total cholesterol (p< 0.3506) ... and serum insulin (p< 0.6502). Page 8- line 43- use past tense—compared with instead of comparing with. Page 9- Line 46 and 59 - use past tense for compare, not present continuous, the data were already compared. Discussion Page 10- Lines 15, 35 and 60 - use past tense for compare, not present continuous, the data were already compared. Page 11- line 17 - use past tense for compare, not present continuous, the data were already compared. Lines 38- 43- Re write “ Third, we included patients who were admitted to this hospital for the first time but it is unclear if they were first diagnosed with T2DM or HTN which made the study population more heterogenous.” References- outdated, only 21 cited for such an important topic. Need to remove the justification and do left justify to fix extra spaces in references. Better formatting needed. Tables and Figures. Need clear titles for figures
--	---

VERSION 1 – AUTHOR RESPONSE

Reviewer: 1

Mr. Gebiso Roba Debele, Mettu University

Comments to the Author:

Yan Liu, Jie Li, Ying Dou and Hongshan Ma; performed a Cohort study, in order to assess the Impacts of type 2 diabetes mellitus and hypertension on cardiovascular disease and stroke in Chinese patients.

The paper is quite original if related to the geographic area. The methodology is correct. The results support the conclusions.

However, this reviewer raises some minor criticisms that have to be addressed by the authors and presented as follows:

General Comments

Abstract

Introduction: Please add the gaps of your study before your expectation

Response: Thanks for your suggestion. We added "However, there is little evidence from large-scale study to assess the joint effect of T2DM and HTN on the risk of cardiovascular events in China."

Methods: Better to add the study design and analysis method

Response: Thanks for your suggestion. We revised according to the Journal's Instructions for Authors: <http://bmjopen.bmj.com/pages/authors/#research>.

Finding: "Findings For the current study, 8,012 patients with T2DM, 9,653 patients with HTN and 3,592 patients with both T2DM" and what??

Response: Thanks for your suggestion. We have revised to "For the current study, 8,012 patients with T2DM only, 9,653 patients with HTN only and 3,592 patients with both T2DM and HTN and 10,561 patients without T2DM or HTN were selected from the SuValue database."

Better to replace the word "Interpretation" by Conclusion in the abstract.

Response: Thanks for your suggestion. We revised to "conclusions".

Methods

"We examined 3 outcomes of interest" but what exists on your manuscript is only 2 outcomes which is VE and stroke. What is your third outcome?

Response: Thanks for your suggestion. We revised to "We examined two outcomes of interest: combined vascular event (VE) and stroke. Combined VE include stroke, myocardial infarction, coronary heart disease, heart failure and coronary bypass, percutaneous coronary intervention."

Where is the operational definition of your outcome?

Response: Thanks for your suggestion. We revised to "The outcomes were defined as the first event or last record before December 31 2019 in the event-free cases according to the diagnosis in the patients' EMRs."

Results

“Prevalent HTN was significantly associated with a 524.6% increased risk of VE (95% CI 5.712-6.830) and 764.2% increased risk of stroke”. What does mean by 524.6% and 764.2% percent? And you need to clarify all those percent's above 100%.

Response: Thanks for your suggestion. We revised to “HTN only group had more than 6-fold increased risk of VE (95% CI 5.712-6.830) and more than 8-fold increased risk of stroke (95% CI 7.517-9.934) in unadjusted models (all Ps<0.0001).”

Some of your references was too old and you need to update.

Response: Thanks for your suggestion. We have updated the references.

Reviewer: 2

Dr. Faraja Chiwanga, Muhimbili National Hospital

Comments to the Author:

General comments:

1) There are several typographical errors throughout the manuscript which makes it difficult to follow arguments the authors are making

Response: Thanks for your suggestion. We revised the manuscript for readability.

2) This study has been described as cross-sectional however there are several facts that indicate it is not a cross-sectional type of study, like; i)excluding patients with outcome of interest and ii)prospectively following up patients to observe occurrence of interest. It is therefore difficult to focus on the results and analysis without first understanding the method the authors used.

Response: Thanks for your suggestion. We did not describe precisely. We have revised to “This was a retrospective, cohort study designed to evaluate the risk of cardiovascular diseases and stroke in patients with T2DM and/or HTN from 2004 to 2015 in China real-world setting.” “Patients were included if they met the following criteria: (1) aged≥18; (2) newly diagnosed T2DM and/or HTN; (3) had the baseline examination records before or within 3 months at the first diagnosis (for details see the Section Baseline parameters); (4) EMRs could be found in one year later after the first diagnosis of T2DM and/or HTN. Patients were excluded as follows: sex information missing; had been diagnosed with stroke, myocardial infarction, coronary heart disease, heart failure, had received coronary artery bypass grafting or percutaneous coronary intervention before the first diagnosis of T2DM and/or HTN.” Besides, we deleted the “All patients were followed up for about 5 years” and “follow-up” in the “outcome and follow-up” section.

Reviewer: 3

Dr. Mario Iannaccone, Ospedale San Giovanni Bosco

Comments to the Author:

The authors should be congratulated for their efforts, in particular they collected a big amount of data. However the paper suffers of important limitation, such as the lack of data about BMI, further the endpoint definition is really unclear (in particular for the composite endpoint).

Response: Thanks for your suggestion. When patients admitted to hospital, doctors do not measure his/her weight and height. Therefore, we could not obtain the information.

We revised the endpoint definition to “We examined two outcomes of interest: combined vascular event (VE) and stroke. Combined VE include stroke, myocardial infarction, coronary heart disease, heart failure and coronary bypass, percutaneous coronary intervention. The outcomes were defined as the first event or last record before December 31. 2019 in the event-free cases according to the diagnosis in the patients’ EMRs.”

Indeed the results/conclusion mainly confirm previous known experience such as the FRAMINGAM study cohort and similar epidemiological studies.

Response: Thanks for your suggestion. We referred to related references in the Discussion section. We added “To our knowledge, our study firstly investigated the combined effect of HTN and T2DM in large cohorts in real world setting in Chinese patients.”

Reviewer: 4

Dr. Sheila Barrett, Northern Illinois University

Comments to the Author:

General- some typos and grammatical mistakes noted. Need to left justify to correct the additional spaces in the document. Need to use the past tense, the study has already been completed. Statistical analyses were rigorous but in the results 3 sentences started off with Kaplan-Meier Analyses, change these beginning sentences.

Response: Thanks for your suggestion. We polished the manuscript for readability.

We revised to “Kaplan–Meier analyses demonstrated increased combined VE and stroke risk in the HTN group.”

“Kaplan–Meier analyses demonstrated increased combined VE and stroke risk in the HTN group.”

“Kaplan–Meier analyses demonstrated increased combined VE and stroke risk in the HTN and T2DM group.”

Abstract

Some typos noted;

Line 9- change was largely increased to “has” largely increased. The study was performed “to” “investigate”

Response: Thanks for your suggestion. We revised to “The prevalence of type 2 diabetes mellitus (T2DM) and hypertension (HTN) has largely increased in recent years.” “This study was performed to investigate the association of T2DM and HTN with the incidence of combined vascular event (VE) and stroke in China.”

Methods- identify the type of study in the abstract, seems like a retrospective cross-sectional study.

Response: Thanks for your suggestion. We revised to “Design A retrospective cohort study.”

Line 22- is confusing, if the study is retrospective, why mention follow -up for 5 years?

Response: Thanks for your suggestion. We deleted “All patients were followed up for about 5 years”

That sounds like a prospective cohort study.

Response: Thanks for your suggestion. We revised to “Design A retrospective cohort study.”

Line 27- add HTN after T2DM

Response: Thanks for your suggestion. We revised to “both T2DM and HTN”.

Interpretation – this should be conclusion?

Response: Thanks for your suggestion. We revised to “Conclusion”.

Rewrite- “Subjects with T2DM and HTN were strongly associated with VE and stroke risk, however, the HTN only group was more strongly associated with VE and stroke risk compared with the T2DM only group.

Response: Thanks for your suggestion. We revised to “Subjects with T2DM and HTN were strongly associated with combined VE and stroke risk, however, the HTN only group was more strongly associated with combined VE and stroke risk compared with the T2DM only group.”

Page 4- strengths and Limitations- separate the strengths from the limitations.

Response: Thanks for your suggestion. We separated the strengths from the limitations.

Line 17- reword “Data on cardiovascular risk factors...”

Response: Thanks for your suggestion. We revised to “Cardiovascular risk factors were collected in this study.”

Lines 28- 34- need a rewrite- Suggestions “We included patients who were admitted to this hospital for the first time but it is unclear if the patient was first diagnosed as type 2 diabetes mellites or

hypertension which increased the heterogeneity of the study population. “Not clear what is meant on the bullet point for mortality and BMI, are these strengths or limitations? Explain to make them clearer as well as separate strengths from limitations.

Response: Thanks for your suggestion. We revised to “□

Limitations of this study

- □ BMI and life style such as smoking and alcohol drinking were not accessed through electronic medical records in hospitals.
- □ Mortality data were also not accessed through the electronic medical records in hospitals.

Introduction

Line 28- cardiovascular diseases “compared” with

Response: Thanks for your suggestion. We have revised to “According to the Framingham Heart Study, adults with diabetes had absolute 2-fold risk of cardiovascular diseases compared with subjects without diabetes.”

Line 35- study showed HTN is commonly “found” in ...

Response: Thanks for your suggestion. We have revised to “A cross-sectional study showed that HTN was commonly in newly diagnosed diabetes, and HTN patients had a higher prevalence of cardiovascular events than normotensive subjects before the diagnosis of diabetes.”

Line 38- add the word “who” after HTN patients

Response: Thanks for your suggestion. We have revised to “A cross-sectional study showed that HTN was commonly in newly diagnosed diabetes, and HTN patients had a higher prevalence of cardiovascular events than normotensive subjects before the diagnosis of diabetes.”

Line 54- edit comparing to “compared” (You are mixing up tenses, present continuous with past tense), the study is over, so use past tense.

Response: Thanks for your suggestion. We deleted the sentence due to the content and we have revised all the “comparing” to the “compared”.

Lines 14-20- fix the purpose of the study. Suggestions “ The purpose of the study was to evaluate the impact of HTN and t2DM on cardiovascular diseases risk and stroke in Chinese adults using the SuValue database.

Response: Thanks for your suggestion. We revised to “The purpose of the study was to evaluate the impact of HTN and T2DM on the risk of cardiovascular disease and stroke in the Chinese adults using the SuValue database.”

Methods

Page 6- Line 33- add “events/diseases after the risk of cardiovascular...”

Response: Thanks for your suggestion. We revised to “This was a retrospective, cohort study designed to evaluate the risk of cardiovascular diseases and stroke in patients with T2DM and/or HTN from 2004 to 2015 in China real-world setting.”

Page 7- line 17- change was to were.

Response: Thanks for your suggestion. We revised to “both ethics consideration and written informed patient consent were not needed for this analysis.”

Page 7- Line 46- remove the superscript “st’ after December 31, superscript is typically used before the month, should remain as December 31, 2019. Or 31st December 2019.

Response: Thanks for your suggestion. We revised to “The outcomes were defined as the first event or last record before December 31, 2019.”

Results

Page 8- line 28- add “the” in front of SuValue

Response: Thanks for your suggestion. We revised to “from the SuValue database”.

Page 8- line 36- add the p values in parentheses - ... for total cholesterol ($p < 0.3506$) ... and serum insulin ($p < 0.6502$).

Response: Thanks for your suggestion. We revised to “Comparing baseline characteristics between the four groups revealed significant differences except for total cholesterol ($p = 0.3506$) and serum insulin ($p = 0.6502$).”

Page 8- line 43- use past tense—compared with instead of comparing with.

Response: Thanks for your suggestion. We revised to “In unadjusted models, compared with non-HTN and non-T2DM patients, the HR of T2DM was 1.747 for combined VE.”

Page 9- Line 46 and 59 - use past tense for compare, not present continuous, the data were already compared.

Response: Thanks for your suggestion. We revised to “Compared with HTN only group, unadjusted HR for combined VE risk was 0.789 and 0.693 for stroke risk in the both T2DM and HTN group”.

“there was no significant reduced risk for both T2DM and HTN group compared with HTN only group.”

Discussion

Page 10- Lines 15, 35 and 60 - use past tense for compare, not present continuous, the data were already compared.

Response: Thanks for your suggestion. We revised to “In this present study, having HTN and/or T2DM was significantly associated with combined VE and stroke before and after adjustment for major cardiovascular risk factors compared with the non-T2DM and non-HTN group.”

“In unadjusted analyses and adjustment model for major cardiovascular risk factors, we observed that patients with both T2DM and HTN showed increased risk of combined VE and stroke compared with those only with T2DM.”

“Thus, combined T2DM and HTN did not increase the risk of combined VE and stroke compared with HTN, which may be due to the antihypertensive treatment in this population.”

Page 11- line 17 - use past tense for compare, not present continuous, the data were already compared.

Response: Thanks for your suggestion. We revised to “A study performed in Iranian older adults showed that T2DM alone increased the all-cause mortality by 62% compared with HTN alone.”

Lines 38- 43- Re write “ Third, we included patients who were admitted to this hospital for the first time but it is unclear if they were first diagnosed with T2DM or HTN which made the study population more heterogenous.”

Response: Thanks for your suggestion. We deleted the sentence due to the improper description.

References- outdated, only 21 cited for such an important topic. Need to remove the justification and do left justify to fix extra spaces in references. Better formatting needed.

Response: Thanks for your suggestion. We added some references and format the references according to Journal’s requirement.

Tables and Figures.

Need clear titles for figures

Response: Thanks for your suggestion. We added the figure legends “Figure 1 Kaplan-Meier survival curve of combined vascular event (A) and stroke (B) among different groups. HTN: hypertension, T2DM: type 2 diabetes mellitus.”